# DIVERSE PREFERENCE OPTIMIZATION

## ABSTRACT

Post-training of language models, either through reinforcement learning, preference optimization or supervised finetuning, tends to sharpen the output probability distribution and reduce the diversity of generated responses. This is particularly a problem for creative generative tasks where varied responses are desired. In this work we introduce *Diverse Preference Optimization (DivPO)*, an optimization method which learns to generate much more diverse responses than standard pipelines, while maintaining the quality of the generations. In DivPO, preference pairs are selected by first considering *a pool* of responses, and a measure of diversity among them, and selecting chosen examples as being more rare but high quality, while rejected examples are more common, but low quality. DivPO results in generating 45.6% more diverse persona attributes, and a 74.6% increase in story diversity while maintaining similar win rates as standard baselines. On general instruction following, DivPO results in a 46.2% increase in diversity, and a 2.4% winrate improvement compared to DPO.

## 1 INTRODUCTION

Large language models (LLMs) are proficient at producing high quality "human-aligned" outputs in response to a particular prompt (Touvron et al., 2023; Gemini et al., 2023; Achiam et al., 2023; Jiang et al., 2024). However, this alignment unfortunately results in a difficulty in producing a diverse set of outputs (Zhang et al., 2024). For example, repeatedly prompting a current state-of-the-art model to write a story with a particular title ends up producing stories with remarkably similar characters, events, and style. Aside from being an issue for individual user queries as just described, this also impacts the ability to generate high quality synthetic data – which is becoming a vital component of model training via AI feedback, where generated data from the model is fed back into the training loop, allowing for self-improvement (Wang et al., 2022b; Bai et al., 2022b; Yuan et al., 2024; Singh et al., 2023; Chen et al., 2024).

The convergence of responses into a limited support distribution appears to stem from the model alignment phase, where the base language model is fine-tuned to align with human outputs and preferences Kirk et al. (2024); Bronnec et al. (2024); Santurkar et al. (2023). Model weights are tuned to optimize a reward (typically a proxy for human preferences). This results in the model placing a high probability on the highest rewarded response, and low on everything else. However, there may be other responses that have the same reward, but are ignored by the training loss. Ideally, we want responses with the same rewards to have the same probability of being generated. Further, when there is a small gap in reward between two responses, we also want their probabilities to be close.

To address this limitation, we propose a novel training method called *Diverse Preference Optimization (DivPO)* which aims to balance the distribution of quality responses given a prompt. The key intuition is that rather than contrasting the highest and lowest rewarded responses as typically done in preference optimization, we instead select the most diverse response that meets a reward (quality) threshold and contrast it with least diverse response that is below a reward threshold. Our method is designed to not only achieve high rewards, reflecting alignment with human preferences, but also to maintain a high degree of diversity among the generated outputs. This dual objective is key for applications where both quality and variety are critical.

We experiment on two creative generation categories: structured synthetic personas, and unstructured creative writing. We first demonstrate how standard optimization approaches in state-of-the-art models increase reward but significantly reduce diversity. We then show how in contrast our method simultaneously increases the reward and the diversity compared to the baseline model. Our method is general and allows the use of any diversity criterion. In particular, we effectively generalize reward models to the case of assigning scores *given a pool of examples*, rather than being pointwise with

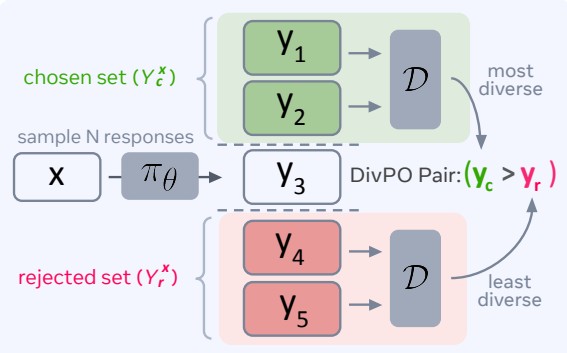

Figure 1: **Diverse Preference Optimization** (DivPO). We consider a diversity criterion $\mathcal{D}$ for selecting chosen and rejected responses from a pool in preference optimization. Rather than taking the highest rewarded response as the chosen ($y_c$), we select the *most diverse* response that meets a certain quality reward threshold. Similarly, the *least diverse* response that is below a threshold is selected as the rejected response ($y_r$). These are contrasted against each other to optimize both quality and diversity simultaneously.

independent scores. We demonstrate that for any fixed quality target, DivPO has higher diversity metrics than any baseline method. We find that on a general instruction following task, DivPO not only leads to more diverse responses, but also higher quality compared to baselines.

## 2 THE ALIGNMENT COLLAPSE PROBLEM

During the pre-training stage, language models are trained with a cross-entropy loss on diverse text corpora, resulting in a model that learns a distribution matching such data (Brown et al., 2020; Touvron et al., 2023). However, the ability of modern language models to respond to instructions in a manner preferred by humans is attributable to a post-training stage where they are "aligned" with human preferences over responses. Consequently, in this reinforcement learning from human feedback (RLHF) (Christiano et al., 2017) stage, the original learned distribution from the pre-training stage collapses. The reason behind this is the objective of reinforcement learning, which aims to optimize the cumulative future reward, $R$: $\mathcal{L} = -\sum_t r_t = -R$, where $r_t$ can be seen as a reward for generating a token at time step $t$ in the case of text generation. Putting all sequence-level probability mass on the highest reward point is an optimal solution to this loss. Even if multiple generations have the same reward, shifting all probability to only one of them is an optimal solution. Therefore, the objective of optimizing the cumulative reward causes collapse.

To combat this, a regularizing KL term with respect to a reference model is often added to the loss:

$$\mathcal{L} = -\sum_t r_t - \beta \text{KL}(\pi || \pi_{\text{ref}})$$

This loss is used in both the PPO (Schulman et al., 2017; Ouyang et al., 2022) and DPO (Rafailov et al., 2024) methods. The optimal solution to this loss is

$$\pi^* \propto \pi_{\text{ref}} \exp(R/\beta).$$

This is more suitable because putting all probability on the highest reward is no longer optimal. However, there can still be collapse depending on $\beta$. Lower $\beta$ means higher reward generations can have even higher probabilities, thus leading to less diversity. We cannot increase $\beta$ freely because it controls the KL term and higher $\beta$ will force the model to stay similar to the original model, which is less well aligned to human preferences. Yet despite the addition of this KL constraint, many works have showed that RLHF substantially decreases the diversity of language model outputs along several linguistic axes (Bai et al., 2022a; Kirk et al., 2024; Guo et al., 2024; Murthy et al., 2024), even when collaborating with humans (Padmakumar & He, 2024).

Furthermore, language models are typically evaluated on the "quality" of their responses, which ignores issues of generator collapse. Common evaluation metrics such as accuracy, pass@N (Kulal et al., 2019; Chen et al., 2021), and win rate (Li et al., 2023; Bai et al., 2022a) measure how frequently

a language model's responses answer a query correctly, pass unit tests, or exhibit higher quality than another model's responses. These metrics can often be optimized even if the model always generates the same response for a given prompt, so long as the response is high quality. However, there are many tasks that benefit from *both* high-quality and diverse responses. Tasks such as creative writing, idea generation, and biological sequence design, explicitly require and benefit from diverse generations (Marco et al., 2024; Tachibana et al., 2025; Madani et al., 2023). Further, recent work on inference scaling laws shows that LLM reasoning improves when searching a more diverse candidate space (Wang et al., 2022a; Yao et al., 2024). In addition, since LLMs are often used to generate synthetic training data, homogenous outputs can lead to significant downstream consequences, such as systemic bias (Yu et al., 2023) and model collapse (Feng et al., 2024; Shumailov et al., 2024).

## 3  METHOD

Given the issues with existing preference optimization methods, we set forth two goals. First, we want high reward generations to be more likely than low reward generations, as is standard. Second, we want all high reward generations to have similar probabilities under the language model distribution. Hence we introduce our method, called Diverse Preference Optimization (DivPO), which aims to optimize both of these goals simultaneously.

In methods like DPO Rafailov et al. (2024), the goal is to optimize the reward margin, so the highest rewarded response is selected as the "chosen" response, and is typically contrasted with the least rewarded response, referred to as the "rejected"[1] (Yuan et al., 2024; Xu et al., 2023; Pace et al., 2024).

In DivPO, we add a second constraint for selecting the chosen and rejected responses. Rather than selecting the highest rewarded response for the chosen, we want the *most diverse* response that meets a certain reward threshold. Similarly, we want to reject the *least diverse* response that is below a reward threshold. We loosely define a response as being more "diverse" if it differs substantially from other responses generated by the same model. We provide a more precise definition of diversity in the sections below. Importantly, our method can accept any threshold and diversity criterion, which lets the user select their reward threshold tolerance and diversity measure that they want to optimize.

Given this constraint, the steps for selecting a training pair for each training prompt are as follows. Given training prompt $x$, we first sample $N$ responses from initial model $\pi_\theta$ to create a pool of candidates, $Y^x = \{y_1, y_2, ..., y_N\}$. We then score each response $y_i$ using a reward model, $s_i = \text{RM}(x, y_i)$. We then establish two buckets by thresholding the reward scores with a hyperparamter $\rho$: the chosen set $Y_c^x$ and rejected set $Y_r^x$, detailed in Section 3.1.

Next, we require diversity criterion (reward) $\mathcal{D}$ which in the general case takes as input a pool of responses $Y$ and outputs a diversity score for each: $d_i = \mathcal{D}(y_i, Y)$. To determine the chosen response, we select the *most diverse* response within the chosen set: $y_c = \text{argmax}_{y_i \in Y_c^x} \mathcal{D}(y_i, Y_c^x)$. To determine the rejected response, we select the *least diverse* response within the rejected set: $y_r = \text{argmin}_{y_i \in Y_r^x} \mathcal{D}(y_i, Y_r^x)$. We introduce several diversity criteria in Section 3.1.

We repeat this process for each training prompt to create a training set of preference pairs, as summarized in Alg. 1. We use this set of diverse chosen and rejected responses to fit a Bradley-Terry model and update model $\pi_\theta$:

$$\mathcal{L}_{\text{DivPO}}(x; \pi_\theta, \pi_{\text{ref}}) = -\log \sigma \left( \beta \log \frac{\pi_\theta(y_c \mid x)}{\pi_{\text{ref}}(y_c \mid x)} - \beta \log \frac{\pi_\theta(y_r \mid x)}{\pi_{\text{ref}}(y_r \mid x)} \right). \quad (1)$$

Here $\beta$ is used to control the deviation from a reference model $\pi_{\text{ref}}$ (we use $\beta$=0.1 for all DivPO experiments). In other words, we introduce a new preference pair selection method, and use the same optimizer as used in direct preference optimization Rafailov et al. (2024).

### 3.1  DIVPO CONFIGURATIONS

**Reward Threshold $\rho$.** To determine the chosen set $Y_c^x$ and rejected set $Y_r^x$, we introduce a hyperparameter $\rho$, which represents the percentage range from the lowest to the highest reward value. All responses that have a reward value within $\rho$ percentage below the highest reward will be added to the chosen set. And all responses that have a reward value within $\rho$ percentage above the lowest reward

---

[1]Other methods such as best-vs-random are also possible, but we use the common best-vs-worst approach.

will be added to the rejected set. In other words, if $\rho = 0$, then the chosen and rejected responses will be identical to DPO, and if $\rho=0.5$, then all responses are considered.

**Diversity Criterion $\mathcal{D}$.** While our algorithm is general enough to allow for any diversity criterion $\mathcal{D}$, we use three different methods to determine the most and least diverse from a set of responses.

- **Model Probability.** If a response $y_i$ has a higher probability under the model, that means it is more likely to be generated again, hence less diverse. Thus we define $\mathcal{D}(y_i) = -\log \pi_\theta(y_i|x)$ so that less likely responses are considered more diverse. Note that this metric does not require a set of responses as input.

- **Word Frequency.** Given a set of responses, we can measure how frequently a specific word occurs. A response with more frequent words is likely to be similar to other responses sharing the same words. Given this, we define $\mathcal{D}$ as inverse word frequency.

- **LLM-as-a-Diversity-Judge.** Finally, in general one could learn or predict diversity from a trained model, similar to reward modeling for quality. We prompt a language model to select the most and least diverse responses from the chosen and reject sets. See Appendix Figure 7 for the prompt.

## 3.2 DivPO Training

DivPO can be used in both offline (off-policy) and online (on-policy) training. For online training the for loop in Alg. 1 is executed at every training step, but only over the current batch of prompts. Compared to an offline setup, online training in other optimization approaches has shown performance improvements (Qi et al., 2024; Noukhovitch et al., 2024) at the cost of computational efficiency. In standard methods however online training is known to be more prone to collapse because as the model generation becomes less diverse, simultaneously so does the model's response training data. Our experiments include both offline and online setups which confirm the effectiveness of DivPO regardless of the chosen training regime.

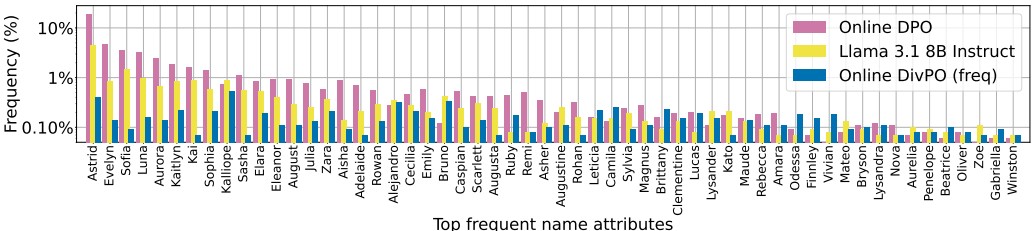

Figure 2: **Persona Generation Statistics**. Llama 3.1-8B-Instruct and DPO tend to repeatedly generate a small subset of names, as shown by frequency (%) of the top most frequently generated. In contrast, DivPO provides a substantially more uniform distribution over the most frequent attributes, in addition to overall improved diversity metrics (see Table 1).

## 4 Experiments

We use the Llama 3.1-8B-Instruct model Dubey et al. (2024) as the baseline model and as initialization checkpoint for training experiments. We utilize the fairseq2 library (Balioglu, 2023) to implement DivPO objective and execute supervised and preference fine-tuning recipes. We utilize the vLLM library (Kwon et al., 2023) as the inference engine for data generation and evaluation. We use NVIDIA H100 GPUs for training and evaluation. Training hyperparameters can be found in Table 6.

### 4.1 Persona Generation Task

**Task.** Our first task aims at generating a random character/person description by prompting the model to produce a JSON object containing various attributes. This is hence a constrained, structured output (with JSON specific keys). Specifically, we choose 3 attributes: first name, city of birth, and occupation. Figure 4 displays the prompt we use.

**Reward.** We define a rule-based reward based on the validity of the JSON output that can be used during training: a reward of 1 is assigned if the model's output contains a valid JSON object (and nothing else) featuring all the required attributes; otherwise a reward of 0 is assigned.

Table 1: **Persona Generation Task Results.** We report diversity and quality metrics comparing different training methods. Diversity for each attribute is defined as the number of unique attributes divided by the total number of generations satisfying the rule-based reward. Generation quality is defined using mean ArmoRM score and the percentage of valid JSON objects. DivPO dramatically improves the diversity of attributes while maintaining the quality. Bold selections highlights best values independently for offline and online training rows.

| Method | Diversity ↑ | | | | Quality ↑ | |
|---|---|---|---|---|---|---|
| | First name | City | Occupation | Avg | ArmoRM | Valid JSON % |
| Llama 3.1-8B-Instruct | 30.45% | 13.82% | 27.93% | 24.07% | 0.141 | 45.51% |
| GPT-4o | 2.55% | 0.74% | 0.38% | 1.22% | 0.140 | 100.00% |
| SFT | 22.18% | 9.14% | 20.98% | 17.43% | **0.142** | **99.58%** |
| DPO | 22.95% | 10.44% | 27.92% | 20.44% | **0.142** | 99.25% |
| DivPO, $\mathcal{D}$=Freq | **49.68%** | 27.87% | 57.47% | 45.01% | 0.139 | 98.73% |
| DivPO, $\mathcal{D}$=Prob | 48.89% | **29.86%** | **58.44%** | **45.73%** | 0.139 | 97.32% |
| Online SFT | 24.92% | 6.92% | 19.46% | 17.10% | 0.139 | 99.54% |
| Online DPO | 11.61% | 3.19% | 10.82% | 8.54% | 0.139 | **99.99%** |
| Online DivPO, $\mathcal{D}$=Freq | 52.04% | **45.35%** | **65.03%** | **54.14%** | **0.141** | 99.80% |
| Online DivPO, $\mathcal{D}$=Prob | **53.85%** | 29.21% | 55.77% | 46.28% | 0.134 | 98.26% |

**Evaluation metrics.** We evaluate models by generating 10000 outputs conditioned on the prompt (ancestral sampling with temperature 1.0), and measure several metrics. We compute the *diversity* of each attribute as the number of unique attributes divided by the total number of generations satisfying the rule-based reward. The *quality* of the model's output is measured by the average score produced by the ArmoRM reward model (Wang et al., 2024b) over the outputs satisfying the rule-based reward. ArmoRM takes as input the prompt and a model's response, and outputs a scalar value indicating response quality. Finally, we report the average rule-based JSON reward as another quality metric.

**Diversity criterion.** When considering diversity of responses, we only consider the pool of valid JSON outputs. Since all valid responses have the same reward of 1, we thus simply make the chosen and rejected pools equal to the all valid responses. We experiment with using both the Probability and Word Frequency criteria for $\mathcal{D}$, as described in Section 3.1. For Frequency, we choose one attribute at random (out of name, city, occupation), and compute its frequency statistics over the valid responses. For Probability we use the length-normalized log probability to pick the chosen and rejected responses. In both settings we also include extra preference pairs that feature random valid and invalid outputs (reward 1 and 0) as chosen and rejected targets in order to alleviate quality degradation during training.

**Preference training.** In this task, we train models using both offline and online regimes. We do checkpoint selection for final evaluation using the rule-based reward value computed over a set of generations every 50 steps. We report checkpoint steps of each training run in Table 5.

**Baselines.** We consider two baseline methods: supervised finetuning (SFT) with NLL loss and preference finetuning using the DPO objective (Rafailov et al., 2024). SFT uses Llama's generated outputs with reward 1 as targets, while DPO uses them as chosen and reward 0 as rejected.

**Results.** While this task appears simple, standard state-of-the-art models such as Llama 3.1-8B-Instruct and GPT-4o fail to produce diverse personas (Table 1). As shown in Figure 2, the Llama 3.1 model generates the name "Astrid" more than 10% of the time, and in general is heavily skewed toward only a few names, as shown in Appendix Figure 12. Adapting the sampling temperature does not alleviate this problem, as we see a sharp decline in quality, shown in Appendix Figure 11.

In contrast, we find our newly proposed method substantially increases diversity in generated personas in both online and offline training regimes, while maintaining quality. As shown in Table 1, online DivPO (using Word Frequency) achieves an average (across attributes) improvement in diversity of up to $30.07\%$ compared to the Instruct model and up to $45.6\%$ compared to online DPO while achieving better quality in that case. The online DivPO training regime shows up to a $9.14\%$ diversity improvement compared to its offline version. We find that both Diversity criteria (Frequency and Probability) achieve similarly strong performance, with slight wins for each in different settings.

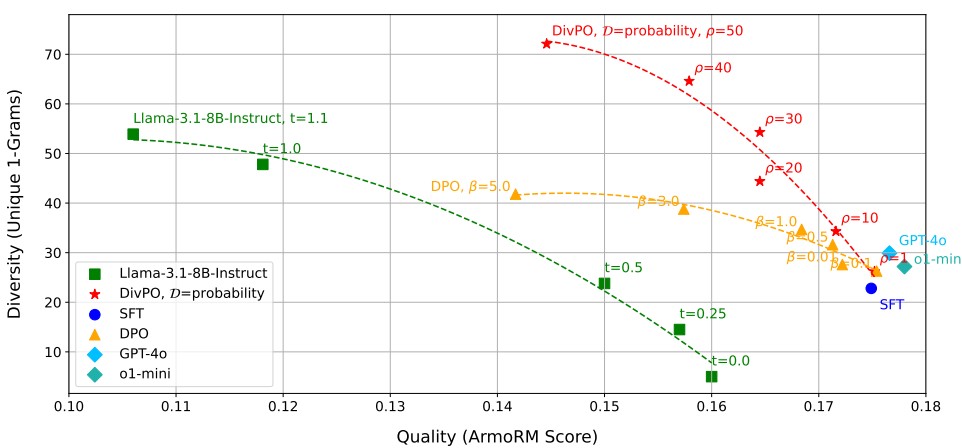

Figure 3: **Keyword Story Generation Results.** We show quality (ArmoRM scores) vs Diversity (Unique 1-Grams) for $N$=16 responses per prompt. To tune diversity for baselines, we vary the baseline Lama 3.1-8B-Instruct temperature ($t$), and the DPO $\beta$ value. For our method, DivPO, we vary the $\rho$ hyperparamter for choosing preference pairs. Unless otherwise noted, all methods use $t$=1.

In addition to generating more unique persona attributes, DivPO improves the distribution over the most frequently generated attributes. Figure 2 presents a histogram of the most frequently generated attributes compared to the Instruct model and DPO. The more uniform frequency distribution of DivPO confirms that it helps to reshape the entire distribution rather than just lengthening its tail.

As reported previously in the literature, we confirm that DPO training is prone to diversity collapse. We also find that this effect is more severe in the online DPO setting compared to the offline one. However, online training shows greater stability during training as seen from both rule-based and ArmoRM reward measured during the training, with more detail shown in Appendix Figure 13. We speculate that this might be related to the task's prompt distribution featuring a single prompt that makes learning saturate much faster during offline training compared to the online version. Overall, DivPO works well in both offline and online cases, with stronger performance in the online setting.

## 4.2 Keyword Story Generation Task

**Task.** Many open ended prompts do not have a predefined set of valid outputs, and require a relatively unstructured creative output. We therefore consider the setting of creative story writing. In this task we thus prompt the language model to write five keywords that could be used in a story with a particular title, as detailed in Figure 5. Here, there is no specific set of valid outputs, but a requirement of exactly five words. We chose the five word story task for several reasons. First, its simple nature allows for systematic anlysis of our method in open ended generation. Second, restricting the output to five words allows us to easily measure diversity, which is non-trivial in open ended tasks (Tevet & Berant, 2020). Since there is no verifiable reward, we rely on a trained reward model to determine response quality. We create the dataset by first generating 6,000 synthetic story titles using Llama 3.1-405B-Instruct (Dubey et al., 2024). Each of these are used in our prompt template (Figure 5) to build our dataset prompts. We take 5k for training and 1k for testing. We then generate $N$=100 responses per training prompt with Llama3.1-8B-Instruct, which are used to create the training preference pairs (or SFT responses).

**Reward.** Following the Persona Generation task, we use the ArmoRM reward model to score responses. During evaluations, we manually set the reward value to 0 if the five word constraint is not met in the response since we observe that ArmoRM does not penalize missing the constraint.

**Evaluation metrics.** For evaluation, we use three diversity metrics: compression ratio, unique 1-grams, and entropy. Details for how the metrics are computed are described in Section C.1. All diversity metrics are computed on the first five words. For quality, we report both the absolute ArmoRM score, as well as the ArmoRM win rate compared to the baseline Llama model.

**Diversity criterion.** For DivPO, we use the three diversity criteria outlined in Section 3.1 (Probability-based, Frequency-based, or LLM-as-a-Diversity-Judge), and reward threshold values

$\rho$ ranging from 0.01 to 0.5. For the Probability-based criterion, we use the log probability of all response tokens. For the frequency-based criterion, we first compute the frequency of each word in each set, $Y_c^x$ and $Y_r^x$, then compute the mean frequency of all words in each response $y_i$. We use the Llama 3.1-405B-Instruct model for the LLM-Judge case.

**Preference training.** Each model is trained for 1k steps, with offline training (more efficient as we require LLM reward model calls). We use temperature=1 for all training and evaluation responses.

**Baselines.** Similar to the Persona task, we consider two baseline training setups. Supervised finetuning (SFT) with NLL loss and preference finetuning using DPO. For each prompt, the top ArmoRM scored response is used for SFT, and best-vs-worst scored responses are used for DPO. We report varying both the quality/diversity tradeoff changing the $\beta$ parameter, and by varying decoding temperature. We also evaluate GPT-4o (Hurst et al., 2024) and o1-mini (Jaech et al., 2024).

**Results.** Figure 3 shows the results for the DivPO ($\mathcal{D}$=Prob) model compared to baselines, evaluating quality and diversity using mean ArmoRM reward scores and unique 1-grams, respectively. GPT-4o, o1-mini, and our SFT and DPO trained models all increase the quality (reward) compared to the baseline Llama 3.1-8B-Instruct model, but decrease the diversity significantly. DivPO similarly increases the quality compared to the base model, but can drastically increase the diversity compared to all baseline models. We show that we can control the amount of diversity or quality by changing the $\rho$ parameter in the DivPO preference pair selection. The DivPO ($\rho = 0.3$) model achieves a 13.6% increase in diversity and 39.6% increase in quality compared to the Llama model. Compared to DPO, DivPO with $\rho = 0.3$ is 74.6% more diverse, with only a drop of 6.4% in quality. Importantly, DivPO models always have *equal or higher diversity at specific quality values* compared to the baselines.

We show the full results for all three diversity criteria $\mathcal{D}$={Probability, Frequency, LLM-Judge}, and and all three diversity evaluation metrics (compression ratio, unique 1-grams, entropy) in Appendix Table 4. DivPO is more diverse and higher quality compared to the baseline Llama 3.1-8B-Instruct model for any diversity criterion. The $\mathcal{D}$=Prob model performs the best with a 35.1% increase in unique 1-grams compared to the Llama 3.1-8B-Instruct model at a higher quality level. $\mathcal{D}$=Word Frequency and $\mathcal{D}$=LLM-Judge increase unique 1-grams by 15.7% and 2%, respectively, at a better quality level than the baseline model. Furthermore, for all criteria, we see a smooth tradeoff between diversity and quality by varying the $\rho$ parameter, demonstrating that whichever diversity criterion users prefer, they can change the diversity/quality threshold to achieve the right level of diversity.

Appendix Figure 8 shows an example test set prompt and the word count (overlap) statistics from the DPO and DivPO ($\mathcal{D}$=Prob, $\rho = 0.3$) models on $N$=16 generations for the story title *"The Eyes of the World"*. The DPO model has a small number of unique words, and a highly skewed distribution within those. The words "witness", "global" and "perspective" dominate the 16 responses. The DivPO model has almost double the total amount of unique words, and a more uniform distribution among them. Both models have a similar mean reward among the 16 generations, indicating that DivPO learned a set of quality, yet diverse responses. We show the full set of responses in Figure 9.

**Full Story Generation Task.** Following the keyword stories experiments, we explore a full story generation task. Rather than the simpler task of generating a summary (5 keywords) in the previous section. To do this, we utilize the generated keywords as seeds for crafting full stories. For each story title, we use $N$=16 keyword story generations derived from a given model. For DivPO, we use the $\mathcal{D}$=Prob models' seeds. Each of these keyword stories serves as a prompt for the Llama 3.1-8B-Instruct model, which then composes a full paragraph story, resulting in 16 stories per title. The template prompt to generate stories is provided in Appendix Figure 6. We evaluate the stories with the same diversity metrics used in the keyword story generation task. We use the ArmoRM model to evaluate story quality using the prompt without the keyword specification (i.e. only `Write a 1 paragraph story with the title {title}`).

Results are shown in Table 3. We find a similar tradeoff between quality and diversity as in the previous task when varying the DivPO $\rho$ parameter. Around $\rho = 0.1$, we see similar quality between DPO and SFT, but higher diversity for DivPO. For $\rho > 0.1$, diversity improves further over the base model, with a slight drop in quality. We show an example of generated stories in Appendix Figure 10.

Table 2: Instruction Following Results. We show both diveristy and quality results for the AlpacaEval 2.0 benchmark, which measures general instruction following capabilities. DivPO is both more diverse and higher quality than the baseline Llama 3.1-8b-instruct model and DPO finetuning.

| Method | Diversity ↑ | | | Quality ↑ | | | |
| | Compr. Ratio | Unique 1-Gram | Entropy | LC Winrate | Winrate | Std Err | Length |
|---|---|---|---|---|---|---|---|
| Llama 3.1-8b-instruct | 0.2497 | 1393.0 | 315.7 | 23.56 | 25.19 | 1.29 | 2643 |
| DPO | 0.1894 | 994.7 | 127.0 | 41.30 | 38.83 | 1.42 | 1915 |
| DivPO, $\mathcal{D}$=Prob, $\rho$=0.10 | 0.2207 | 1181.3 | 202.2 | 42.68 | 41.99 | 1.44 | 1978 |
| DivPO, $\mathcal{D}$=Prob, $\rho$=0.20 | 0.2247 | 1177.5 | 189.0 | **44.07** | 39.66 | 1.44 | 1835 |
| DivPO, $\mathcal{D}$=Prob, $\rho$=0.30 | 0.2464 | 1326.6 | 242.6 | 41.70 | 41.41 | 1.46 | 1985 |
| DivPO, $\mathcal{D}$=Prob, $\rho$=0.40 | 0.2598 | 1454.6 | 284.1 | 42.29 | **42.50** | 1.46 | 2138 |
| DivPO, $\mathcal{D}$=Prob, $\rho$=0.50 | **0.2949** | **1909.5** | **506.4** | 31.31 | 32.82 | 1.39 | 3878 |

### 4.3 INSTRUCTION FOLLOWING TASK

**Task.** Lastly, we explore the effectiveness of using DivPO on general instruction following tasks. We train on a random set of 10k single-turn Wildchat prompts (Zhao et al., 2024), which is an open source dataset of 1 million real-world user-ChatGPT interactions.

**Evaluation metrics.** We evaluate using the AlpacaEval 2.0 benchmark (Dubois et al., 2024; Li et al., 2023), which uses GPT-4 as a judge to compute winrates against GPT4-turbo. This benchmark evaluates general instruction following capabilities on a diverse set of tasks. We sample 32 times for each prompt to compute diversity metrics (same as story generation). We then select 1 sample per prompt for AlpacaEval tests.

**Diversity criterion.** We use the Probability-based diversity criterion (since it is simple and shown effective in the toy task experiments), and reward threshold values $\rho$ ranging from 0.01 to 0.5.

**Preference training.** Similar to story generation, we utilize the ArmoRM model to create preference pairs from $N = 32$ samples per prompt. We train each model for a fixed 500 steps.

**Baselines.** We compare to the seed Llama 3.1-8b-instruct model and vanilla DPO.

**Results.** Results are shown in Table 2. We find that DivPO for all $\rho$ values between 0.10 and 0.40 are more diverse and higher quality than DPO. Additionally, for $\rho = 0.40$, the DivPO responses are significantly higher quality than the baseline Llama 3.1-8b-instruct model, with a similar or greater diversity (greater compression ratio and unique 1-gram diversity).

The motivation of our method is to mitigate the diversity collapse that occurs when training with vanilla DPO. We can see that while DPO improves the quality compared to the baseline model, the diversity drops drastically. DivPO, however does not have the same collapse, yet still improves the quality compared to both the baseline as well as the DPO model. Our results indicate that training with a diversity constraint can not only mitigate collapse, but in fact lead to higher quality responses at test time because the model learns a wider variety of quality responses during training. These experiments demonstrate a promising result that the quality/diversity tradeoff does not necessarily have to hold when using DivPO; we can simultaneously increase both quality and diversity.

For this task, we also include ablations of the N (samples per prompt) hyperparameter during training on the 10k Wildchat samples, shown in Table 7. We ablate 3 different values of $N = \{16, 32, 64\}$ (samples per prompt) during training. For simplicity, we measure quality using ArmoRM on a set of 470 heldout prompts: 253 valid set examples from Humpback (Li et al. 2024) and 218 examples from the Evol-Test. (Xu et al. 2023). All evaluations are on 32 samples per prompt in the test set. We find that our method is robust to to all three $N$ sizes, where we show that DivPO (particularly for $\rho = \{0.1, 0.2\}$) outperforms DPO on all metrics for all $N$ values. We find that higher values of $\rho$ work better on smaller $N$.

## 5 RELATED WORK

**Preference Optimization and Collapse.** Reinforcement Learning from Human Feedback (RLHF) is a crucial component of training LLMs that align with human preferences (Ouyang et al., 2022).

DPO (Rafailov et al., 2024) and other preference optimization methods (Xu et al., 2023; Meng et al., 2024) have significantly simplified the RLHF process and yield similar improvements. While these methods improve performance and generalization they can also negatively affect diversity and calibration (Achiam et al., 2023; Kirk et al., 2024). In particular, RLHF methods optimize the final reward which does not take diversity into account, so it has become common practice to add a KL regularization term to maintain some of the model's original diversity (Ziegler et al., 2019; Rafailov et al., 2024). Wang et al. (2023) find that RLHF under reverse KL divergence regularization can express a limited range of political views. Wang et al. (2023) claim that the drop in diversity is because of the KL term. They test various f-divergences with DPO and show that the reverse KL gives the best accuracy but worst diversity, while forward KL gives best diversity but sacrifices accuracy.

**Mitigating Collapse During Training.** To promote diversity, Li et al. (2015) propose an alternative to SFT called Maximum Mutual Information, where they seek to maximize the pairwise mutual information between the source and the target. Similarly, Li et al. (2024) introduce a entropy-regularized alternative to cross-entropy in order to mitigate SFT collapse. Zhang et al. (2024) consider structured output tasks where they know an ideal distribution and introduce a supervised loss to match the model distribution with the ideal distribution. Yu et al. (2024) propose an iterative refinement method to resample instances from clusters during training. Cideron et al. (2024) include both CFG distillation and a diversity reward (the cosine similarity of embeddings) trained with RL, and apply it to music generation. Bradley et al. (2023) propose an evolutionary algorithm to discover unique (based on Pugh et al. (2016)), yet quality responses and measure quality using a binary language model output. Chung et al. (2023) propose label replacement to correct misaligned labels in the training data. Padmakumar et al. (2024) propose a new method for collecting human preference data that encourages diversity. Lastly, several modifications to DPO were proposed to increase diversity. Wang et al. (2024a) include multiple chosen and multiple rejected samples when creating preference pairs, but do not sample all responses from the same model or use an explicit reward model to allow for online self-improvement. Park et al. (2024) augment DPO with an online diversity sampling term for maximizing protein structure stability and sequence diversity. Qin et al. (2025) propose a method for iterative self-improvement which re-uses previous model generations and improves the diversity of reasoning chains without compromising final answer accuracy. To the best of our knowledge, we introduce the first method which directly modifies the preference pair selection process in order to simultaneously optimize quality and diversity.

**Eliciting Heterogeneous Outputs During Inference.** The simplest method for eliciting more or less heterogeneous outputs from a pretrained language model is to change the sampling distribution (Holtzman et al., 2019; Fan et al., 2018; Nguyen et al., 2024; Dhuliawala et al., 2024). Zhang et al. (2020) demonstrate the tradeoff between diversity and quality using sampling hyperparamters such as temperature, top-k, and top-p, and find that increasing all three perform similarly in order to increase diversity. However, sampling higher temperatures is known to lead to nonsensical generations (Tevet & Berant, 2020). Eldan & Li (2023) enforce diversity by conditioning generations to incorporate randomly chosen words into a story, but this may not be applicable for all writing prompts.

## 6 CONCLUSION

In this paper, we introduced Diverse Preference Optimization (DivPO), a novel training method designed to address the lack of diversity in current state-of-the-art model responses, which is at least partially caused by existing preference optimization techniques. Our approach aims to add variety and balance the distribution of high quality responses by promoting diversity while maintaining alignment with high quality human preferences.

We demonstrated that standard optimization methods, while effective at increasing reward, often lead to a significant reduction in output diversity. In contrast, DivPO successfully enhances both quality (reward) and diversity, as evidenced by multiple creative generation tasks, as well as general instruction following.

Our method allows for users to define their own diversity criterion to be optimized, as well as a hyperparameter to control the diversity vs. quality tradeoff. DivPO can be directly plugged into existing preference optimization frameworks, allowing users to easily achieve their target levels of output diversity. The main limitation of our method is that it requires a preference optimization loss, and does not work for other RLHF losses such as GRPO. Future work could extend our method to further tasks, and integrate new methods of measuring diversity and rewards.

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

---

**Algorithm 1** DivPO Preference Pair Creation

---

**Require:** Training set $T$, diversity criterion $\mathcal{D}$, reward threshold $\rho$, base model $\pi_\theta$, reward model $RM$,

**for** prompt $x$ in $T$: **do**

    1. Sample $N$ responses: $\{y_1, y_2, ..., y_N\} \sim \pi_\theta(x)$

    2. Score each response: $s_i = RM(y_i, x)$

    3. Determine chosen set $Y_c^x$ and rejected set $Y_r^x$ based on reward values and threshold $\rho$.

    4. Use diversity criterion to find most diverse from the chosen set: $y_c = \text{argmax}_{y_i \in Y_c^x} \mathcal{D}(y_i, Y_c^x)$.

    Use diversity criterion to find least diverse from the rejected set: $y_r = \text{argmin}_{y_i \in Y_r^x} \mathcal{D}(y_i, Y_r^x)$.

    5. Add $(x, y_c, y_r)$ to set of preference pairs

**end for**

---

```
Persona generation

Generate a random persona description with three characteristics.\n\n
Characteristics are:\n\n
- First Name\n
- The city of birth\n
- Current occupation\n\n
Format the output strictly using JSON schema. Use `first_name` for First Name,
`city` for the city of birth, `occupation` for current occupation as
corresponding JSON keys. The ordering of characteristics should be arbitrary
in your answer.
```

Figure 4: Instruction for the structured persona generation task.

```
Keyword Story Generation

List 5 words that could be used in a story titled "{title}".
Do not write anything else but a list of 5 words without numbers.
```

Figure 5: Instruction template for the keyword story generation task.

# A STATEMENTS

## A.1 REPRODUCIBILITY STATEMENT

We will provide code required to reproduce our results on the benchmark datasets. In the manuscript, we include the type of GPUs that we require to reproduce our experiments. We also provided standard error values for reproducing our results across different seeds.

## A.2 ETHICS STATEMENT

We conform to the ICLR Code of Ethics at https://iclr.cc/public/CodeOfEthics.

## A.3 LLM USAGE STATEMENT

We did not use LLMs for ideation or writing.

# B DIVPO DETAILS

Algorithm 1 outlines a detailed description of how to create DivPO preference pairs for training.

---

**Full Story Generation**

```
Write a 1 paragraph story with the title "{title}".
Your story must include the following elements: "{keywords}".
```

Figure 6: Instruction template for the full story generation task.

---

**LLM Diversity Criterion Prompt**

```
You will be given a list of phrases, where each phrase is specified by an index
number writtten as "[PHRASE_INDEX]". Select the index of the MOST UNIQUE phrase
compared to all other phrases. Consider unique words, word rarity, phrase
structure, etc. Do not explain your answer, just write the index.

Phrases:
{top_response_list}

Write your answer in the following format: "[PHRASE_INDEX]".
```

Figure 7: Prompt for selecting the most diverse keyword story out of a list of stories. A similar prompt is used for selecting the least diverse story, by replacing MOST UNIQUE with LEAST UNIQUE.

## C    METRICS

### C.1    DIVERSITY IN STORY GENERATION TASKS

A diversity metric $M_d$, takes a set of model responses $Y$, and produces a scalar score of how diverse the set is Kirk et al. (2024).

Given a model, $\pi$, we retrieve a set of $N$ outputs from the model for each prompt $x$ in the test prompt set $T$:

$$Y_\pi^x := \{y_i \sim \pi(y \mid x) \mid i = 1, \ldots, N\}$$

For a diversity metric $M_d$, we then evaluate the Prompt Level Diversity (the diversity of $Y_\pi^x$)

$$\text{Prompt Level Diversity} := \frac{1}{|T|} \sum_{x \in T} M_d(Y_\pi^x). \tag{2}$$

For all metrics, we compute the prompt level diversity for the set of $N$ responses, and then average over all prompts in the test set.

**Compression Ratio.** Shaib et al. (2024) show that compression ratio (CR) is a surprisingly strong diversity metric that is more robust than other commonly used metrics. To compute compression rate for N responses, we first concatenate them into a single sequence, $[y_1, y_2, \ldots y_N]$, then compress it using gzip. Finally, we compute the byte size ratio of the compressed sequence and original sequence:

$$\text{Compression Ratio} = \frac{\text{SIZE}(\text{gzip}([y_1, y_2, \ldots y_N]))}{\text{SIZE}([y_1, y_2, \ldots y_N])} \tag{3}$$

**Unique 1-Grams.** Unique 1-grams counts the number of distinct n-grams in the set of outputs. Since the tasks we consider in this paper are short sequences, we do not utilize $n > 1$ grams.

**Entropy.** Lastly, we consider the entropy of the responses, which can be estimated from the set of responses $\{y_1, y_2, \ldots y_N\}$:

$$\text{Entropy}_\pi(x) = - \sum_y \pi(y|x) \log \pi(y|x) \tag{4}$$

$$= \mathbb{E}_{y \sim \pi(y|x)}[-\log \pi(y|x)] \tag{5}$$

$$\approx \frac{1}{N} \sum_{i=1}^{N} [-\log \pi(y_i|x)] \tag{6}$$

Table 3: **Full Story Generation Results.** We use the keywords generated by each model in the "Method" column as seeds for a Llama 3.1-8B-Instruct model to generate a full paragraph story. For DivPO we use the $\mathcal{D}$=Prob model. Similar to the original keyword stories task, we observe a trend that DivPO has higher diversity, while maintaining similar quality compared to the baseline Llama 3.1-8B-Instruct model.

| Method | Diversity ↑ | | | Quality ↑ |
|---|---|---|---|---|
| | Compr. Ratio | Unique 1-Gram | Entropy | ArmoRM |
| Llama3.1-8B-Inst. | 0.3663 | 765.6 | 193.0 | 0.1609 |
| SFT | 0.3510 | 704.4 | 188.3 | 0.1633 |
| DPO, $\beta$=0.1 | 0.3525 | 708.1 | 187.8 | 0.1638 |
| DivPO, $\rho$=0.01 | 0.3530 | 709.4 | 187.5 | 0.1638 |
| DivPO, $\rho$=0.1 | 0.3601 | 736.7 | 193.3 | 0.1625 |
| DivPO, $\rho$=0.2 | 0.3698 | 783.9 | 206.4 | 0.1603 |
| DivPO, $\rho$=0.3 | 0.3767 | 814.4 | 208.3 | 0.1600 |
| DivPO, $\rho$=0.4 | 0.3855 | 844.2 | 216.2 | 0.1581 |
| DivPO, $\rho$=0.5 | 0.3981 | 921.1 | 242.8 | 0.1528 |

## C.2 QUALITY

In the structured tasks, we have an explicitly defined measurement of response quality. If the response is a valid JSON output, then the quality/reward is 1, and if the response isn't a valid JSON, then the reward is 0.

In the unstructured tasks, there is no easily verifiable measurement of quality. We therefore use the ArmoRM reward model Wang et al. (2024b). This model takes as input a user response $x$ and a response $y$ and outputs a scalar value $r$ representing the quality of the response with respect to the prompt. In the keyword stories task, we manually set the reward to 0 if the word length constraint is not met.

## D ADDITIONAL KEYWORD STORY EXPERIMENTS

Table 4 shows the full experimental results for the keywords stories tasks. We observe generalization of our DivPO across different types of diversity criteria, $\mathcal{D}$.

## E ADDITIONAL PERSONA GENERATION TASK RESULTS

Figure 12 shows the distribution of generated name attributes of personas for two different temperature settings (1.0 and 1.4) of Llama 3.1-8B-Instruct on the Persona Generation Task. We see that the higher temperature generates personas more uniformly.

However, in Figure 11 we show the diversity vs. quality trade-off as we change the sampling temperature in the Persona Generation Task. We demonstrate that as we increase temperature, diversity increases, but quality (measured by ArmoRM score) decreases.

## F ADDITIONAL INSTRUCTION FOLLOWING EXPERIMENTS

In Table 7 we ablate the number of generations $N$ for the instruction following task. We find that DivPO is robust to varying $N$ values, and outperforms DPO for all values.

**DPO. Unique Words: 31, Mean ArmoRM Score: 0.1512**

witness (11), global (10), vigilant (6), gaze (5), vision (4), watchful (4), observer (4), omni (3), panoramic (3), observant (3), observation (2), perspective (2), insight (2), universal (2), panopticon (1), sight (1), omnipresent (1), aware (1), view (1), cosmology (1), spectacle (1), panorama (1), mysterious (1), surveillance (1), all-seeing (1), far-reaching (1), insights (1), all (1), voyeur (1), cosmopolitan (1), mystery (1)

**DivPO (ours). Unique Words: 58, Mean ArmoRM Score: 0.1512**

omniscient (5), ominous (4), visionary (3), vigilant (3), prophecy (3), spectacle (3), enigmatic (3), panoramic (2), maelstrom (2), panopticon (2), mirage (2), mystical (2), perspicacious (1), legacy (1), riddle (1), ethereal (1), luminary (1), horizon (1), alluring (1), mysterious (1), biblical (1), fathomless (1), vastness (1), inscrutable (1), surveillance (1), amulet (1), omens (1), renaissance (1), aesthete (1), mirrored (1), obsidian (1), providence (1), intrigue (1), pioneering (1), sunlit (1), venerable (1), sentinel (1), pinnacle (1), prophetic (1), inescapable (1), unified (1), clarity (1), luminous (1), witness (1), unveiling (1), transcendent (1), observed (1), farsighted (1), voyeur (1), vast (1), spectacular (1), destiny (1), wistful (1), cosmopolitan (1), ennui (1), meditative (1), cerebral (1), universality (1)

Figure 8: **Keyword Stories Example Statistics**. We show word count statistics for $N$=16 generations of both the DPO and DivPO ($\rho$=0.3, $\mathcal{D}$=Prob) model responses for the story title *"The Eyes of the World"*. The DPO model has a small number of unique words, and a highly skewed distribution. DivPO, has the same quality (ArmoRM score) as DPO with nearly double the amount of unique words, and a more uniform distribution among them.

Table 4: Keyword Stories Results. We show the mean metric values for the $N$=16 responses per prompt. Winrates are compared to the baseline Llama 3.1-8B-Instruct model. We demonstrate three different diversity criteria: probability, word count, and LLM. DivPO is both more diverse and achieves higher ArmoRM scores compared to the baseline model. DivPO is significantly more diverse than DPO, while maintaining similar winrates.

| | Diversity ↑ | | | Quality ↑ | |
|---|---|---|---|---|---|
| **Method** | **Compr. Ratio** | **Unique 1-Gram** | **Entropy** | **ArmoRM** | **ArmoRM Winrate** |
| Llama 3.1-8B-Instruct | 0.498 | 47.8 | 20.3 | 0.1178 | - |
| GPT-4o | 0.326 | 29.9 | - | 0.1766 | 0.8715 |
| o1-mini | 0.361 | 27.2 | - | 0.1780 | 0.8877 |
| SFT | 0.316 | 22.8 | 5.8 | 0.1752 | 0.8290 |
| DPO | 0.396 | 31.1 | 14.0 | 0.1759 | 0.8351 |
| DivPO, $\rho$=0.01, $\mathcal{D}$=Prob | 0.362 | 26.2 | 11.1 | 0.1752 | 0.8787 |
| DivPO, $\rho$=0.1, $\mathcal{D}$=Prob | 0.412 | 34.3 | 15.0 | 0.1716 | 0.8410 |
| DivPO, $\rho$=0.2, $\mathcal{D}$=Prob | 0.461 | 44.4 | 19.9 | 0.1645 | 0.7586 |
| DivPO, $\rho$=0.3, $\mathcal{D}$=Prob | 0.518 | 54.3 | 24.7 | 0.1645 | 0.6774 |
| DivPO, $\rho$=0.4, $\mathcal{D}$=Prob | 0.562 | 64.6 | 31.2 | 0.1579 | 0.5367 |
| DivPO, $\rho$=0.5, $\mathcal{D}$=Prob | 0.597 | 72.1 | 36.8 | 0.1446 | 0.3973 |
| DivPO, $\rho$=0.01, $\mathcal{D}$=Freq | 0.375 | 28.0 | 11.9 | 0.1711 | 0.8560 |
| DivPO, $\rho$=0.1, $\mathcal{D}$=Freq | 0.413 | 34.9 | 14.6 | 0.1691 | 0.8471 |
| DivPO, $\rho$=0.2, $\mathcal{D}$=Freq | 0.458 | 43.0 | 18.2 | 0.1675 | 0.7909 |
| DivPO, $\rho$=0.3, $\mathcal{D}$=Freq | 0.471 | 43.7 | 17.8 | 0.1652 | 0.7230 |
| DivPO, $\rho$=0.4, $\mathcal{D}$=Freq | 0.476 | 45.6 | 18.3 | 0.1654 | 0.6883 |
| DivPO, $\rho$=0.5, $\mathcal{D}$=Freq | 0.540 | 55.3 | 22.5 | 0.1574 | 0.5502 |
| DivPO, $\rho$=0.01, $\mathcal{D}$=LLM | 0.357 | 25.9 | 10.9 | 0.1762 | 0.8840 |
| DivPO, $\rho$=0.1, $\mathcal{D}$=LLM | 0.393 | 31.4 | 13.4 | 0.1575 | 0.7908 |
| DivPO, $\rho$=0.2, $\mathcal{D}$=LLM | 0.428 | 36.7 | 15.4 | 0.1673 | 0.7857 |
| DivPO, $\rho$=0.3, $\mathcal{D}$=LLM | 0.456 | 42.3 | 17.7 | 0.1662 | 0.7167 |
| DivPO, $\rho$=0.4, $\mathcal{D}$=LLM | 0.496 | 48.8 | 20.6 | 0.1622 | 0.5997 |
| DivPO, $\rho$=0.5, $\mathcal{D}$=LLM | 0.543 | 56.9 | 24.4 | 0.1545 | 0.4819 |

Table 5: Model selection (see Section 4.1 for details) steps for each method in the Persona Generation Task.

| Method | Model selection step |
|---|---|
| SFT | 500 |
| DPO | 50 |
| DivPO (freq) | 100 |
| DivPO (prob) | 50 |
| Online SFT | 400 |
| Online DPO | 400 |
| Online DivPO (freq) | 350 |
| Online DivPO (prob) | 50 |

Prompt

List 5 words that could be used in a story titled "The Eyes of the World".

DPO. Unique Words: 31, Mean ArmoRM Score: 0.1512

1. Vision Global Witness Observation Perspective
2. Vision Global Gaze Witness Panopticon
3. Vision Watchful Global Sight Observation
4. Witness Observer Gaze Insight Omnipresent
5. Aware Witness Omni View Global
6. Cosmology Witness Omni Gaze Spectacle
7. Vision Witness Observer Global Panorama
8. Vigilant Witness Global Insight Panoramic
9. Observant Mysterious Gaze Universal Watchful
10. Vigilant Witness Surveillance All-seeing
11. Witness Gaze Vigilant Observer Panoramic
12. Witness Far-reaching Insights Panoramic
13. Observant Vigilant Watchful Global All-
14. Universal Voyeur Witness Global Observant
15. Cosmopolitan Mystery Perspective Global Vigilant
16. Vigilant Observer Watchful Omni Global

DivPO (ours). Unique Words: 58, Mean ArmoRM Score: 0.1512

1. Ominous Perspicacious Legacy Riddle Panoramic
2. Visionary Maelstrom Ethereal Luminary Panopticon
3. Omniscient Vigilant Mirage Horizon Prophecy
4. Alluring Mysterious Biblical Fathomless Vastness
5. Prophecy Inscrutable Surveillance Panoramic Mystical
6. Amulet Vigilant Omens Renaissance Aesthete
7. Omniscient Maelstrom Mirrored Obsidian Providence
8. Ominous Visionary Intrigue Pioneering Spectacle
9. Spectacle Sunlit Venerable Sentinel Omniscient
10. Mystical Pinnacle Prophetic Inescapable Unified
11. Clarity Prophecy Luminous Witness Unveiling
12. Spectacle Transcendent Observed Mirage Vigilant
13. Panopticon Enigmatic Farsighted Voyeur Omniscient
14. Ominous Enigmatic Vast Spectacular Destiny
15. Wistful Cosmopolitan Ennui Visionary Ominous
16. Meditative Enigmatic Omniscient Cerebral Universality

Figure 9: **Keyword Story Example**. Here we show the full set of $N$=16 generations for one of the test prompts on the keyword stories task. DivPO is more diverse, with similar quality compared to DPO.

Table 6: Hyperparameter values used during training.

| Method | Personas | | Keyword Stories | | Instruction Following | |
| | Batch Size | Learning Rate | Batch Size | Learning Rate | Batch Size | Learning Rate |
| --- | --- | --- | --- | --- | --- | --- |
| SFT | 16 | 1e-7 | 64 | 1e-6 | 64 | 1e-6 |
| DPO | 16 | 1e-7 | 64 | 1e-7 | 64 | 1e-7 |
| DivPO | 16 | 1e-7 | 64 | 1e-7 | 64 | 1e-7 |

Prompt

Write a 1 paragraph story with the title "The City of Lost Sounds". Your story must include the following elements: "{keywords}".

DPO. Unique Words: 675 Mean ArmoRM Score: 0.1800

1. In the City of Lost Sounds, the air vibrated with a multitude of melodies that refused to settle on a single harmony. Echoes of forgotten symphonies lingered in abandoned alleys, their haunting strains blending with the whispers of long-forgotten secrets on the wind. Meanwhile, a fugue of clashing notes wrestled through the city's main square, as ancient churches competed with modern skyscrapers in a battle for sonic supremacy. Yet, in the midst of this aural chaos, Silentium stood watch, a guardian of the city's hidden corners where the cacophony of everyday life receded, and the Listener's Journey began. It was said that only those who tuned their ears to the whispers of Silence could uncover the truth hidden within the city's dissonant fabric.
2. In the City of Lost Sounds, echoes of forgotten melodies lingered in the streets, where whispers of yesterday's conversations seemed to fade away with the setting sun. Murmurs of distant laughter and forgotten gossip swirled through the crowded alleys, but the city's inhabitants had grown accustomed to tuning them out. Among the skyscrapers, silhouettes of people went about their daily lives, their shadows cast on the pavement like fleeting shadows on a forgotten photograph. It was a city where sound was a rare and precious commodity, and those who dwelled within its boundaries felt the weight of silence as a physical force, a reminder that even the most profound moments of human connection can, with time, become nothing more than a distant memory.
3. "The City of Lost Sounds" In the City of Lost Sounds, the air was alive with ephemeral whispers that danced on the wind, only to fade away like wisps of smoke. It was a place where melodies once sweet and clear began to shatter and break, leaving only faint echoes that struggled to resurface. As I walked through the city's deserted streets, I couldn't help but feel a sense of longing for the sounds that had once filled the air. Fading recollections of laughter and music lingered, a haunting reminder of what had been lost. The city's very fabric seemed to absorb the sounds, leaving only an aching silence that sought to steal what remained of the melodies. And yet, in the city's hidden places, I began to hear the faintest of murmurs - whispers that refused to be silenced, straining to rise up and reclaim their rightful place in the sonic tapestry of the city, an eternal search for the elusive beauty that had once resided there.
4. In the city of Lost Sounds, the air was heavy with the echoes of forgotten melodies. Whispers of forgotten songs lingered in the alleys, where rainwater had warped the pavement into twisted, melodic ridges. On main streets, the cacophony of car horns and chattering pedestrians overwhelmed, threatening to consume the remnants of a long-abandoned symphony. Yet, in secret gardens and hidden courtyards, silence reigned, where the sounds of the city were muffled by ivy and stone. Those who sought refuge in these quiet oases could almost hear the whispers of a long-forgotten composer, their notes tauntingly out of reach, yet still echoing through the desolate beauty of the city of Lost Sounds.

DivPO (ours). Unique Words: 802 Mean ArmoRM Score: 0.1750

1. In the City of Lost Sounds, the air was alive with ephemeral rumblings that seemed to emanate from the very fabric of the metropolis itself. Amidst the cacophony of car horns, chatter, and construction, a faint echoicism resonated through the streets, as if the city's own heartbeat was trying to be heard above the din. The city's denizens, attuned to the subtle vibrations, navigated its alleys and thoroughfares with ease, their footsteps creating their own peculiar rhythms that blended with the cacophony. And yet, amidst this sonic tapestry, there existed moments of eerie silence – pockets of resonance that seemed to hold the city's secrets, waiting to be uncovered by those attuned to the whispers of the lost sounds that dwelled within the city's crumbling walls.
2. The City of Lost Sounds was a metropolis where the very fabric of reality was woven from the threads of forgotten melodies. Here, the adherents of Echoism believed that every whispered secret, every muttered complaint, and every lingering note of music was trapped in the city's earthen heartbeat. As night descended, the murmuration of starlings took flight, their iridescent wings beating in unison, creating an aural tapestry that was both haunting and beautiful. But for those who dwelled in the City of Lost Sounds, the most breathtaking sight was the Fugue of Forgotten Songs, where the shattered remnants of melodies coalesced into a cacophonous symphony that seemed to shroud the city in an impenetrable veil. And at the heart of this auditory labyrinth, the Requiem of Lost Tones echoed through eternity, a somber dirge for the silences that had never been sung.
3. In the City of Lost Sounds, the air was alive with whispers of what could never be heard. Amorphous echoes of chirping birds, like the ephemeral tweets of the Chimarera, danced on the edges of perception. Melodies that started as gentle fugue states, only to twist and fragment into discordant notes, echoed through the vacant streets. As dusk descended, the night sky grew dark with the Murmuration of starlings, their rustling wings beating out a celestial rhythm that seemed to seep into the city's very bones. And yet, it was the distant call of the Siren, a haunting, heart-stopping whisper that pierced the twilight like a shard of glass, that summoned the lost souls of sound to the city's forsaken hearts. For in this mystical metropolis, the lost sounds were not only whispers of what might have been, but also a siren's call to those who dared to listen, and the madness that would ensue.
4. **The City of Lost Sounds** In the City of Lost Sounds, whispers of forgotten melodies wafted through the streets, where architectural facades seemed to have been design to harvest the city's perpetual auroral light. The inhabitants, known as The Broadcasters, possessed an uncanny ability to weave sonic tapestries from the ephemeral harmonies that danced on the meridian lines of the city's celestial layout. With a flick of their wrists, they could conjure luminesce interventions that accentuated the chiaroscuro contrasts of the surrounding buildings, imbuing the urban landscape with an otherworldly aura. Amidst the city's tapestry of sounds, one particular Broadcasters' studios stood out for its mastery of Echter harmonies - rare, painful-sweet melodies that could pierce the heart and evoke memories even the most fervent mnemonists couldn't recall. There, the city's most skilled euphonic artisans crafted tapestries that resonated like a fugue, weaving a sonic topography that seemed to synchronize with the very meridian of the listener's being - a symphony that traversed dimensions, leaving an eternal reverb of only one sound: silence.

Figure 10: **Full Story Example**. We show the first 4 stories generated by the DPO and DivPO ($\mathcal{D}$=Prob, $\rho$=0.3) models. We report the Unique Words and Mean ArmoRM reward for all $N = 16$ stories.

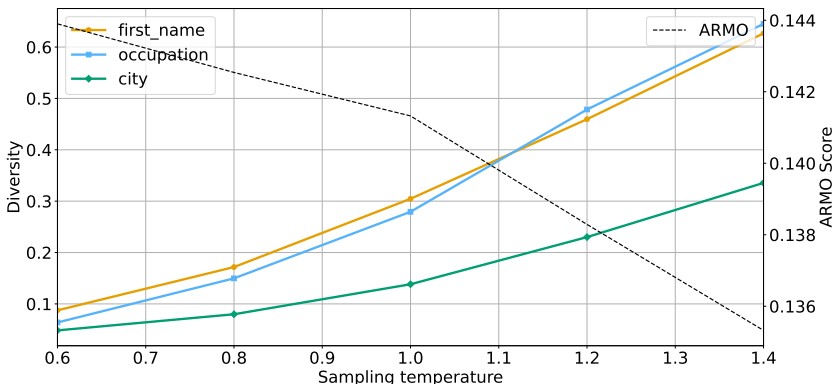

Figure 11: Diversity vs. quality trade-off of the Llama 3.1-8B-Instruct model as we change the sampling temperature in the Persona Generation Task.

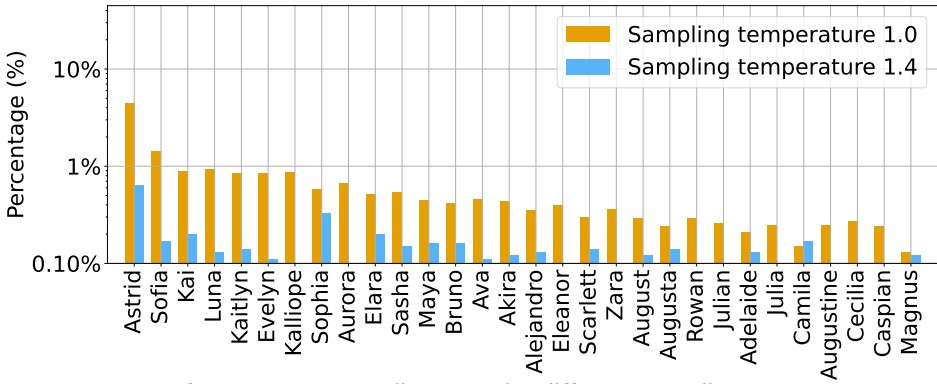

Figure 12: Counts (as percentage) of generated name attributes (computed only over valid JSON outputs) of persona for two temperature settings of Llama 3.1-8B-Instruct on the Persona Generation Task.

Table 7: **Instruction Following Ablations.** Here, we ablate the number of samples per prompt, $N$ during training. We find that DivPO is robust to changes in $N$, across all $\rho$ values.

| Method | N | Compr Ratio ↑ | Unique 1-Grams ↑ | Entropy ↑ | ArmoRM↑ | Len. (words) |
|--------|---|---------------|------------------|-----------|---------|--------------|
| DPO | 16 | 0.1949 | 1115.9 | 165.9 | 0.1803 | 283.4 |
| DivPO, $\mathcal{D}$=Prob, $\rho$=0.1 | 16 | 0.2107 | 1248.4 | 216.7 | 0.1814 | 288.4 |
| DivPO, $\mathcal{D}$=Prob, $\rho$=0.2 | 16 | 0.2187 | 1324.9 | 222.8 | 0.1804 | 285.9 |
| DivPO, $\mathcal{D}$=Prob, $\rho$=0.3 | 16 | 0.2370 | 1460.3 | 274.5 | 0.1794 | 295.0 |
| DivPO, $\mathcal{D}$=Prob, $\rho$=0.4 | 16 | 0.2501 | 1608.2 | 316.2 | 0.1764 | 298.2 |
| DivPO, $\mathcal{D}$=Prob, $\rho$=0.5 | 16 | 0.2632 | 1787.7 | 396.9 | 0.1699 | 301.9 |
| DPO | 32 | 0.1873 | 1110.2 | 146.2 | 0.1792 | 283.9 |
| DivPO, $\mathcal{D}$=Prob, $\rho$=0.1 | 32 | 0.2206 | 1321.0 | 237.7 | 0.1801 | 292.0 |
| DivPO, $\mathcal{D}$=Prob, $\rho$=0.2 | 32 | 0.2232 | 1329.3 | 227.2 | 0.1794 | 280.6 |
| DivPO, $\mathcal{D}$=Prob, $\rho$=0.3 | 32 | 0.2427 | 1478.6 | 282.8 | 0.1780 | 288.7 |
| DivPO, $\mathcal{D}$=Prob, $\rho$=0.4 | 32 | 0.2519 | 1584.5 | 317.6 | 0.1776 | 295.9 |
| DivPO, $\mathcal{D}$=Prob, $\rho$=0.5 | 32 | 0.2851 | 2003.1 | 537.9 | 0.1620 | 304.2 |
| DPO | 64 | 0.1936 | 1121.2 | 163.4 | 0.1791 | 286.3 |
| DivPO, $\mathcal{D}$=Prob, $\rho$=0.1 | 64 | 0.2205 | 1325.6 | 234.2 | 0.1798 | 288.6 |
| DivPO, $\mathcal{D}$=Prob, $\rho$=0.2 | 64 | 0.2395 | 1480.9 | 289.5 | 0.1790 | 296.7 |
| DivPO, $\mathcal{D}$=Prob, $\rho$=0.3 | 64 | 0.2625 | 1760.4 | 414.8 | 0.1698 | 307.3 |
| DivPO, $\mathcal{D}$=Prob, $\rho$=0.4 | 64 | 0.2706 | 1965.8 | 511.6 | 0.1442 | 301.6 |
| DivPO, $\mathcal{D}$=Prob, $\rho$=0.5 | 64 | 0.2965 | 2500.7 | 741.5 | 0.1244 | 316.3 |

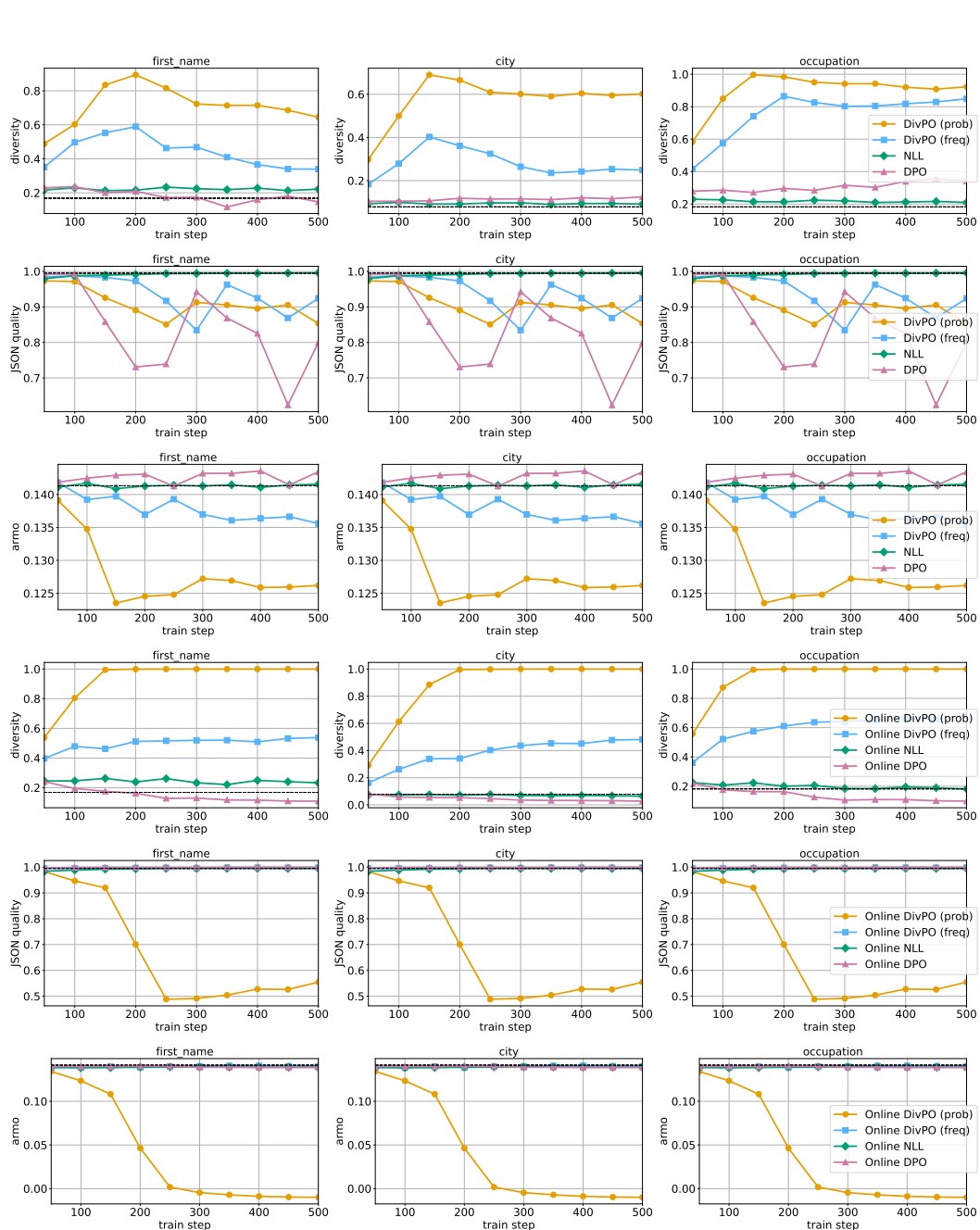

Figure 13: Diversity and quality evaluation in structured persona generation task. Dashed line shows performance of pretrained Llama 3.1 8B Instruct. All generations done with sampling temperature 1.0.

