# OpenReview forum: "Diverse Preference Optimization"
_ICLR.cc/2026/Conference — Submitted to ICLR 2026_

### Official Review · Reviewer_e7H1 · 2025-10-31

**Soundness:** 2
**Presentation:** 2
**Contribution:** 2
**Rating:** 2
**Confidence:** 4

**Summary:**

The paper addresses the diversity collapse problem during post-training of LLMs. The authors propose Diverse Preference Optimization (DivPO), which changes how preference pairs are constructed before applying a standard DPO loss. For each prompt, the model samples a pool of responses, scores them with a reward model, and then forms two sets via a reward threshold. Instead of picking the highest-reward response like standard DPO, DivPO selects the most diverse response from the chosen set and the least diverse response from the rejected set, using one of three diversity criteria: inverse model probability, inverse word frequency, or an LLM-based diversity judge. The authors conduct experiments on structured personas, keyword/full story writing, and instruction following tasks.

**Strengths:**

* The diversity collapse issue is an important and timely problem for creative tasks.
* The proposed idea is simple and can be easily integrated into the existing DPO pipeline.
* The paper explores multiple diversity metrics and evaluates the method across a wide range of tasks.

**Weaknesses:**

* Although the alignment collapse problem is discussed in Section 2, it would be more compelling if the authors provided additional theoretical analysis or more concrete examples/qualitative results illustrating the problem.
* There is prior work addressing a similar problem, and the core idea of this paper appears quite similar to that of [1].
* At the beginning of Section 3, the authors state that ”we want all high reward generations to have similar probabilities under the language model distribution”. However, the main objective of the proposed method does not seem to reflect this goal.
* Determining the optimal threshold and diversity criterion may be challenging, as performance varies considerably across tasks and depends on several factors (dataset and reward distribution).
* Regarding evaluation, it is difficult to verify whether the method truly improves performance, since increased diversity often comes at the cost of reduced output quality (particularly in Tables 1 and 3).

[1] Chung, John Joon Young, et al. "Modifying Large Language Model Post-Training for Diverse Creative Writing." arXiv preprint arXiv:2503.17126 (2025).

**Questions:**

Please see Weaknesses.

---

### Official Review · Reviewer_XygR · 2025-10-31

**Soundness:** 2
**Presentation:** 3
**Contribution:** 2
**Rating:** 4
**Confidence:** 3

**Summary:**

The paper proposes DivPO, which incorporates diversity into the preference comparison sampling process. This approach enhances the diversity of the optimized model’s outputs without compromising performance.

The paper introduce three diversity criterion, model probability, word frequency, and LLM-as-a-diversity-judge. They validate the effectiveness of DivPO on three types of tasks: persona generation, full story generation, and instruction following.

**Strengths:**

The manuscript is clearly written and easy to understand.

The research problem is novel.

**Weaknesses:**

See questions below

**Questions:**

1. The method introduces a hyperparameter ρ. How should ρ be chosen in practice?

2. How generalizable is this approach? Specifically, can the diversity learned from training on Task A transfer to Task B?

3. Does enhancing diversity negatively affect certain tasks, such as mathematical or programming abilities?

---

### Official Review · Reviewer_rSdp · 2025-11-05

**Soundness:** 3
**Presentation:** 3
**Contribution:** 3
**Rating:** 4
**Confidence:** 3

**Summary:**

This paper addresses alignment collapse from DPO by proposing Diverse Preference Optimization (DivPO), a novel training method that optimizes for both quality and diversity simultaneously. DivPO introduces a separate diversity criterion to select the most diverse response from the set of candidate responses chosen by a reward model as high-quality, and the least diverse (most common) response from the low-quality set chosen by the reward model. Then, DivPO trains the model using standard DPO on this new pair. DivPO significantly improves diversity while maintaining similar win rates as DPO and other standard baselines.

**Strengths:**

This paper successfully tackles a practical problem in post-trained LLMs with a simple and intuitive, yet effective method. DivPO intervenes at the data-selection stage, which makes it easy to plug this method into existing DPO pipelines.
The paper shows that DivPO can increase diversity without decreasing quality, which makes the method useful for practical applications.

**Weaknesses:**

The paper currently lacks comparison to other diversity methods as baselines, it would be important to add at least one method that is designed to promote diversity to compare against.
On experiments, there are three tasks studied, but each task experiment uses different choices of diversity criteria. Why not use all three criteria in all experiments?
Also the only realistic task evaluated on is instruction following, but the diversity evaluations seem brittle, so I would love to see diversity evaluated by something like LLM-as-judge or that is more correlated with human diversity perception to be fully convinced of the method's effectiveness.

**Questions:**

What is the computational overhead of DivPO compared to DPO?

---

### Meta-Review · Area_Chair_quhb · 2026-01-07

**Summary:**

This paper introduces Diverse Preference Optimization (DivPO), a plug-and-play variant of DPO that re-weights preference pairs by diversity rather than reward alone, yielding gains in persona, story and instruction and following diversity without hurting win rate. However, the submission raised significant concerns: no baselines from prior diversity promoting methods, three different diversity metrics used across tasks obscure generalizability, brittle automatic diversity metrics that may not reflect human perception, and missing ablation on the sensitive ρ threshold. In the rebuttal phase, the authors did not actively respond to the main concerns. Based on the above considerations, I recommend rejection, while encouraging the authors to further develop this promising direction.

**Reviewer Concerns:**

Still outstanding: all concerns.

**Reviewer Scores:**

All Reviewers: keep negative.

---

### Decision · Program_Chairs · 2026-01-26

Reject